# Assessing Generative Models via Precision and Recall

**Mehdi S. M. Sajjadi**[*]
MPI for Intelligent Systems,
Max Planck ETH Center
for Learning Systems

**Olivier Bachem**
Google Brain

**Mario Lucic**
Google Brain

**Olivier Bousquet**
Google Brain

**Sylvain Gelly**
Google Brain

## Abstract

Recent advances in generative modeling have led to an increased interest in the study of statistical divergences as means of model comparison. Commonly used evaluation methods, such as the Fréchet Inception Distance (FID), correlate well with the perceived quality of samples and are sensitive to mode dropping. However, these metrics are unable to distinguish between different failure cases since they only yield one-dimensional scores. We propose a novel definition of precision and recall for distributions which disentangles the divergence into two separate dimensions. The proposed notion is intuitive, retains desirable properties, and naturally leads to an efficient algorithm that can be used to evaluate generative models. We relate this notion to total variation as well as to recent evaluation metrics such as Inception Score and FID. To demonstrate the practical utility of the proposed approach we perform an empirical study on several variants of Generative Adversarial Networks and Variational Autoencoders. In an extensive set of experiments we show that the proposed metric is able to disentangle the quality of generated samples from the coverage of the target distribution.

## 1 Introduction

Deep generative models, such as Variational Autoencoders (VAE) [12] and Generative Adversarial Networks (GAN) [8], have received a great deal of attention due to their ability to learn complex, high-dimensional distributions. One of the biggest impediments to future research is the lack of quantitative evaluation methods to accurately assess the quality of trained models. Without a proper evaluation metric researchers often need to visually inspect generated samples or resort to qualitative techniques which can be subjective. One of the main difficulties for quantitative assessment lies in the fact that the distribution is only specified implicitly – one can learn to sample from a predefined distribution, but cannot evaluate the likelihood efficiently. In fact, even if likelihood computation were computationally tractable, it might be inadequate and misleading for high-dimensional problems [22].

As a result, surrogate metrics are often used to assess the quality of the trained models. Some proposed measures, such as Inception Score (IS) [20] and Fréchet Inception Distance (FID) [9], have shown promising results in practice. In particular, FID has been shown to be robust to image corruption, it correlates well with the visual fidelity of the samples, and it can be computed on unlabeled data.

However, all of the metrics commonly applied to evaluating generative models share a crucial weakness: Since they yield a one-dimensional score, they are unable to distinguish between different failure cases. For example, the generative models shown in Figure 1 obtain similar FIDs but exhibit

---

[*]This work was done during an internship at Google Brain.
Correspondence: msajjadi.com, bachem@google.com, lucic@google.com.

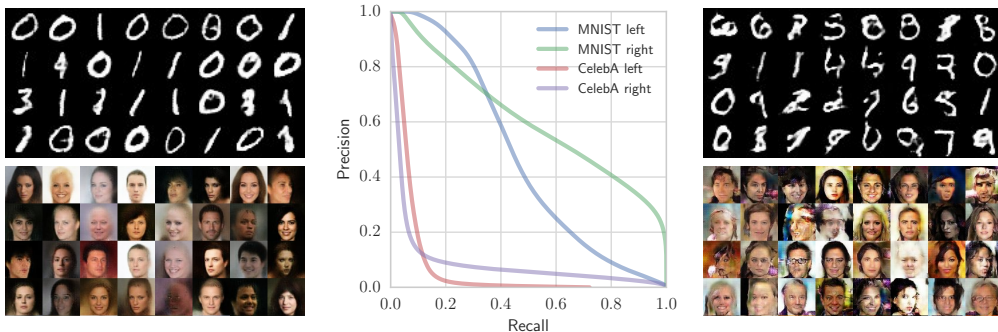

Figure 1: Comparison of GANs trained on MNIST and CelebA. Although the models obtain a similar FID on each data set (32/29 for MNIST and 65/62 for CelebA), their samples look very different. For example, the model on the left produces reasonably looking faces on CelebA, but too many dark images. In contrast, the model on the right produces more artifacts, but more varied images. By the proposed metric (middle), the models on the left achieve higher precision and lower recall than the models on the right, which suffices to successfully distinguishing between the failure cases.

different sample characteristics: the model on the left trained on MNIST [15] produces realistic samples, but only generates a subset of the digits. On the other hand, the model on the right produces low-quality samples which appear to cover all digits. A similar effect can be observed on the CelebA [16] data set. In this work we argue that a single-value summary is not adequate to compare generative models.

Motivated by this shortcoming, we present a novel approach which disentangles the divergence between distributions into two components: *precision* and *recall*. Given a reference distribution $P$ and a learned distribution $Q$, precision intuitively measures the quality of samples from $Q$, while recall measures the proportion of $P$ that is covered by $Q$. Furthermore, we propose an elegant algorithm which can compute these quantities based on samples from $P$ and $Q$. In particular, using this approach we are able to quantify the degree of *mode dropping* and *mode inventing* based on samples from the true and the learned distributions.

**Our contributions:** **(1)** We introduce a novel definition of precision and recall for distributions and prove that the notion is theoretically sound and has desirable properties, **(2)** we propose an efficient algorithm to compute these quantities, **(3)** we relate these notions to total variation, IS and FID, **(4)** we demonstrate that in practice one can quantify the degree of mode dropping and mode inventing on real world data sets (image and text data), and **(5)** we compare several types of generative models based on the proposed approach – to our knowledge, this is the first metric that experimentally confirms the folklore that GANs often produce "sharper" images, but can suffer from mode collapse (high precision, low recall), while VAEs produce "blurry" images, but cover more modes of the distribution (low precision, high recall).

## 2 Background and Related Work

The task of evaluating generative models is an active research area. Here we focus on recent work in the context of deep generative models for image and text data. Classic approaches relying on comparing log-likelihood have received some criticism due the fact that one can achieve high likelihood, but low image quality, and conversely, high-quality images but low likelihood [22]. While the likelihood can be approximated in some settings, kernel density estimation in high-dimensional spaces is extremely challenging [22, 24]. Other failure modes related to density estimation in high-dimensional spaces have been elaborated in [10, 22]. A recent review of popular approaches is presented in [5].

The Inception Score (IS) [20] offers a way to quantitatively evaluate the quality of generated samples in the context of image data. Intuitively, the conditional label distribution $p(y|x)$ of samples containing meaningful objects should have low entropy, while the label distribution over the whole data set $p(y)$ should have high entropy. Formally, $\text{IS}(G) = \exp(\mathbb{E}_{x \sim G}[d_{KL}(p(y|x), p(y))])$. The score is computed based on a classifier (Inception network trained on ImageNet). IS necessitates a labeled data set and has been found to be weak at providing guidance for model comparison [3].

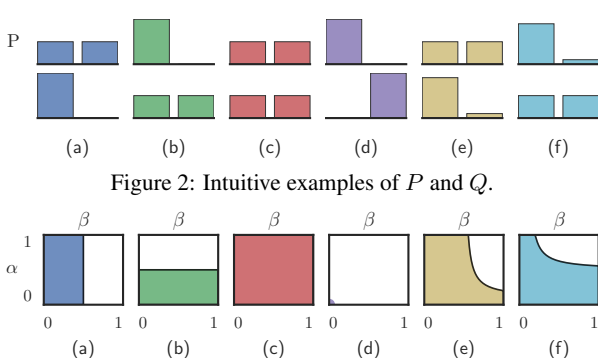
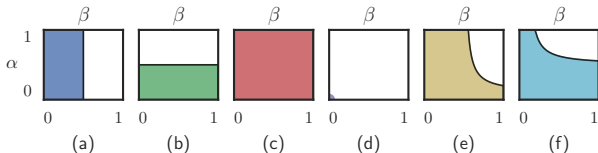
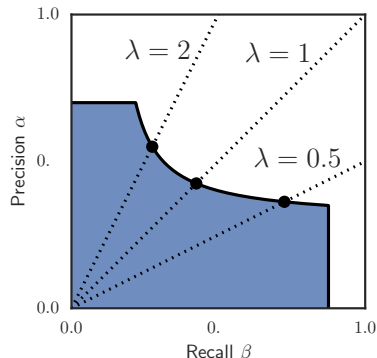

Figure 2: Intuitive examples of $P$ and $Q$.

Figure 3: $\mathrm{PRD}(Q, P)$ for the examples above.

Figure 4: Illustration of the algorithm.

The FID [9] provides an alternative approach which requires no labeled data. The samples are first embedded in some feature space (e.g., a specific layer of Inception network for images). Then, a continuous multivariate Gaussian is fit to the data and the distance computed as $\mathrm{FID}(x, g) = ||\mu_x - \mu_g||_2^2 + \mathrm{Tr}(\Sigma_x + \Sigma_g - 2(\Sigma_x \Sigma_g)^{\frac{1}{2}})$, where $\mu$ and $\Sigma$ denote the mean and covariance of the corresponding samples. FID is sensitive to both the addition of spurious modes as well as to mode dropping (see Figure 5 and results in [18]). [4] recently introduced an unbiased alternative to FID, the *Kernel Inception Distance*. While unbiased, it shares an extremely high Spearman rank-order correlation with FID [14].

Another approach is to train a classifier between the real and fake distributions and to use its accuracy on a test set as a proxy for the quality of the samples [11, 17]. This approach necessitates training of a classifier for each model which is seldom practical. Furthermore, the classifier might detect a single dimension where the true and generated samples differ (e.g., barely visible artifacts in generated images) and enjoy high accuracy, which runs the risk of assigning lower quality to a better model.

To the best of our knowledge, all commonly used metrics for evaluating generative models are one-dimensional in that they only yield a single score or distance. A notion of precision and recall has previously been introduced in [18] where the authors compute the distance to the manifold of the true data and use it as a proxy for precision and recall on a synthetic data set. Unfortunately, it is not possible to compute this quantity for more complex data sets.

## 3 PRD: Precision and Recall for Distributions

In this section, we derive a novel notion of precision and recall to compare a distribution $Q$ to a reference distribution $P$. The key intuition is that *precision* should measure how much of $Q$ can be generated by a "part" of $P$ while *recall* should measure how much of $P$ can be generated by a "part" of $Q$. Figure 2 (a)-(d) show four toy examples for $P$ and $Q$ to visualize this idea: (a) If $P$ is bimodal and $Q$ only captures one of the modes, we should have perfect precision but only limited recall. (b) In the opposite case, we should have perfect recall but only limited precision. (c) If $Q = P$, we should have perfect precision and recall. (d) If the supports of $P$ and $Q$ are disjoint, we should have zero precision and recall.

### 3.1 Derivation

Let $S = \mathrm{supp}(P) \cap \mathrm{supp}(Q)$ be the (non-empty) intersection of the supports[2] of $P$ and $Q$. Then, $P$ may be viewed as a two-component mixture where the first component $P_S$ is a probability distribution on $S$ and the second component $P_{\overline{S}}$ is defined on the complement of $S$. Similarly, $Q$ may be rewritten as a mixture of $Q_S$ and $Q_{\overline{S}}$. More formally, for some $\bar{\alpha}, \bar{\beta} \in (0, 1]$, we define

$$P = \bar{\beta} P_S + (1 - \bar{\beta}) P_{\overline{S}} \quad \text{and} \quad Q = \bar{\alpha} Q_S + (1 - \bar{\alpha}) Q_{\overline{S}}. \tag{1}$$

This decomposition allows for a natural interpretation: $P_{\overline{S}}$ is the part of $P$ that cannot be generated by $Q$, so its mixture weight $1 - \bar{\beta}$ may be viewed as a loss in recall. Similarly, $Q_{\overline{S}}$ is the part of $Q$ that cannot be generated by $P$, so $1 - \bar{\alpha}$ may be regarded as a loss in precision. In the case where

$P_S = Q_S$, i.e., the distributions $P$ and $Q$ agree on $S$ up to scaling, $\bar{\alpha}$ and $\bar{\beta}$ provide us with a simple two-number precision and recall summary satisfying the examples in Figure 2 (a)-(d).

If $P_S \neq Q_S$, we are faced with a conundrum: Should the differences in $P_S$ and $Q_S$ be attributed to losses in precision or recall? Is $Q_S$ inadequately "covering" $P_S$ or is it generating "unnecessary" noise? Inspired by PR curves for binary classification, we propose to resolve this predicament by providing a trade-off between precision and recall instead of a two-number summary for any two distributions $P$ and $Q$. To parametrize this trade-off, we consider a distribution $\mu$ on $S$ that signifies a "true" common component of $P_S$ and $Q_S$ and similarly to (1), we decompose both $P_S$ and $Q_S$ as

$$P_S = \beta'\mu + (1 - \beta')P_\mu \quad \text{and} \quad Q_S = \alpha'\mu + (1 - \alpha')Q_\mu. \tag{2}$$

The distribution $P_S$ is viewed as a two-component mixture where the first component is $\mu$ and the second component $P_\mu$ signifies the part of $P_S$ that is "missed" by $Q_S$ and should thus be considered a recall loss. Similarly, $Q_S$ is decomposed into $\mu$ and the part $Q_\mu$ that signifies noise and should thus be considered a precision loss. As $\mu$ is varied, this leads to a trade-off between precision and recall.

It should be noted that unlike PR curves for binary classification where different thresholds lead to different classifiers, trade-offs between precision and recall here do not constitute different models or distributions – the proposed PRD curves only serve as a description of the characteristics of the model with respect to the target distribution.

## 3.2 Formal definition

For simplicity, we consider distributions $P$ and $Q$ that are defined on a finite state space, though the notion of precision and recall can be extended to arbitrary distributions. By combining (1) and (2), we obtain the following formal definition of precision and recall.

**Definition 1.** *For $\alpha, \beta \in (0, 1]$, the probability distribution $Q$ has precision $\alpha$ at recall $\beta$ w.r.t. $P$ if there exist distributions $\mu$, $\nu_P$ and $\nu_Q$ such that*

$$P = \beta\mu + (1 - \beta)\nu_P \quad \text{and} \quad Q = \alpha\mu + (1 - \alpha)\nu_Q. \tag{3}$$

The component $\nu_P$ denotes the part of $P$ that is "missed" by $Q$ and encompasses both $P_{\overline{S}}$ in (1) and $P_\mu$ in (2). Similarly, $\nu_Q$ denotes the noise part of $Q$ and includes both $Q_{\overline{S}}$ in (1) and $Q_\mu$ in (2).

**Definition 2.** *The set of attainable pairs of precision and recall of a distribution $Q$ w.r.t. a distribution $P$ is denoted by $\mathrm{PRD}(Q, P)$ and it consists of all $(\alpha, \beta)$ satisfying Definition 1 and the pair $(0, 0)$.*

The set $\mathrm{PRD}(Q, P)$ characterizes the above-mentioned trade-off between precision and recall and can be visualized similarly to PR curves in binary classification: Figure 3 (a)-(d) show the set $\mathrm{PRD}(Q, P)$ on a 2D-plot for the examples (a)-(d) in Figure 2. Note how the plot distinguishes between (a) and (b): Any symmetric evaluation method (such as FID) assigns these cases the same score although they are highly different. The interpretation of the set $\mathrm{PRD}(Q, P)$ is further aided by the following set of basic properties which we prove in Section A.1 in the appendix.

**Theorem 1.** *Let $P$ and $Q$ be probability distributions defined on a finite state space $\Omega$. The set $\mathrm{PRD}(Q, P)$ satisfies the following properties:*

$(i) \quad (1, 1) \in \mathrm{PRD}(Q, P) \quad \Leftrightarrow \quad Q = P \hfill \textit{(equality)}$

$(ii) \quad \mathrm{PRD}(Q, P) = \{(0, 0)\} \quad \Leftrightarrow \quad \mathrm{supp}(Q) \cap \mathrm{supp}(P) = \emptyset \hfill \textit{(disjoint supports)}$

$(iii) \quad Q(\mathrm{supp}(P)) = \bar{\alpha} = \max_{(\alpha, \beta) \in \mathrm{PRD}(Q, P)} \alpha \hfill \textit{(max precision)}$

$(iv) \quad P(\mathrm{supp}(Q)) = \bar{\beta} = \max_{(\alpha, \beta) \in \mathrm{PRD}(Q, P)} \beta \hfill \textit{(max recall)}$

$(v) \quad (\alpha', \beta') \in \mathrm{PRD}(Q, P) \text{ if } \alpha' \in (0, \alpha], \beta' \in (0, \beta], (\alpha, \beta) \in \mathrm{PRD}(Q, P) \hfill \textit{(monotonicity)}$

$(vi) \quad (\alpha, \beta) \in \mathrm{PRD}(Q, P) \quad \Leftrightarrow \quad (\beta, \alpha) \in \mathrm{PRD}(P, Q) \hfill \textit{(duality)}$

Property (i) in combination with Property (v) guarantees that $Q = P$ if the set $\mathrm{PRD}(Q, P)$ contains the interior of the unit square, see case (c) in Figures 2 and 3. Similarly, Property (ii) assures that whenever there is no overlap between $P$ and $Q$, $\mathrm{PRD}(Q, P)$ only contains the origin, see case (d) of Figures 2 and 3. Properties (iii) and (iv) provide a connection to the decomposition in (1) and allow an analysis of the cases (a) and (b) in Figures 2 and 3: As expected, $Q$ in (a) achieves a maximum precision of 1 but only a maximum recall of 0.5 while in (b), maximum recall is 1 but maximum

precision is 0.5. Note that the quantities $\bar{\alpha}$ and $\bar{\beta}$ here are by construction the same as in (1). Finally, Property (vi) provides a natural interpretation of precision and recall: The precision of $Q$ w.r.t. $P$ is equal to the recall of $P$ w.r.t. $Q$ and *vice versa*.

Clearly, not all cases are as simple as the examples (a)-(d) in Figures 2 and 3, in particular if $P$ and $Q$ are different on the intersection $S$ of their support. The examples (e) and (f) in Figure 2 and the resulting sets $\mathrm{PRD}(Q, P)$ in Figure 3 illustrate the importance of the trade-off between precision and recall as well as the utility of the set $\mathrm{PRD}(Q, P)$. In both cases, $P$ and $Q$ have the same support while $Q$ has high precision and low recall in case (e) and low precision and high recall in case (f). This is clearly captured by the sets $\mathrm{PRD}(Q, P)$. Intuitively, the examples (e) and (f) may be viewed as noisy versions of the cases (a) and (b) in Figure 2.

### 3.3 Algorithm

Computing the set $\mathrm{PRD}(Q, P)$ based on Definitions 1 and 2 is non-trivial as one has to check whether there exist suitable distributions $\mu$, $\nu_P$ and $\nu_Q$ for all possible values of $\alpha$ and $\beta$. We introduce an equivalent definition of $\mathrm{PRD}(Q, P)$ in Theorem 2 that does not depend on the distributions $\mu$, $\nu_P$ and $\nu_Q$ and that leads to an elegant algorithm to compute practical PRD curves.

**Theorem 2.** *Let $P$ and $Q$ be two probability distributions defined on a finite state space $\Omega$. For $\lambda > 0$ define the functions*

$$\alpha(\lambda) = \sum_{\omega \in \Omega} \min\left(\lambda P(\omega), Q(\omega)\right) \quad and \quad \beta(\lambda) = \sum_{\omega \in \Omega} \min\left(P(\omega), \frac{Q(\omega)}{\lambda}\right). \quad (4)$$

*Then, it holds that*

$$\mathrm{PRD}(Q, P) = \left\{(\theta\alpha(\lambda), \theta\beta(\lambda)) \mid \lambda \in (0, \infty), \theta \in [0, 1]\right\}.$$

We prove the theorem in Section A.2 in the appendix. The key idea of Theorem 2 is illustrated in Figure 4: The set of $\mathrm{PRD}(Q, P)$ may be viewed as a union of segments of the lines $\alpha = \lambda\beta$ over all $\lambda \in (0, \infty)$. Each segment starts at the origin $(0, 0)$ and ends at the maximal achievable value $(\alpha(\lambda), \beta(\lambda))$. This provides a surprisingly simple algorithm to compute $\mathrm{PRD}(Q, P)$ in practice: Simply compute pairs of $\alpha(\lambda)$ and $\beta(\lambda)$ as defined in (4) for an equiangular grid of values of $\lambda$. For a given angular resolution $m \in \mathbb{N}$, we compute

$$\widehat{\mathrm{PRD}}(Q, P) = \{(\alpha(\lambda), \beta(\lambda)) \mid \lambda \in \Lambda\} \quad \text{where} \quad \Lambda = \left\{\tan\left(\frac{i}{m+1}\frac{\pi}{2}\right) \mid i = 1, 2, \ldots, m\right\}.$$

To compare different distributions $Q_i$, one may simply plot their respective PRD curves $\widehat{\mathrm{PRD}}(Q_i, P)$, while an approximation of the full sets $\mathrm{PRD}(Q_i, P)$ may be computed by interpolation between $\widehat{\mathrm{PRD}}(Q_i, P)$ and the origin. An implementation of the algorithm is available at https://github.com/msmsajjadi/precision-recall-distributions.

### 3.4 Connection to total variation distance

Theorem 2 provides a natural interpretation of the proposed approach. For $\lambda = 1$, we have

$$\alpha(1) = \beta(1) = \sum_{\omega \in \Omega} \min\left(P(\omega), Q(\omega)\right) = \sum_{\omega \in \Omega} \left[P(\omega) - (P(\omega) - Q(\omega))^+\right] = 1 - \delta(P, Q)$$

where $\delta(P, Q)$ denotes the total variation distance between $P$ and $Q$. As such, our notion of precision and recall may be viewed as a generalization of total variation distance.

## 4 Application to Deep Generative Models

In this section, we show that the algorithm introduced in Section 3.3 can be readily applied to evaluate precision and recall of deep generative models. In practice, access to $P$ and $Q$ is given via samples $\hat{P} \sim P$ and $\hat{Q} \sim Q$. Given that both $P$ and $Q$ are continuous distributions, the probability of generating a point sampled from $Q$ is 0. Furthermore, there is strong empirical evidence that comparing samples in image space runs the risk of assigning higher quality to a worse model [17, 20, 22]. A common remedy is to apply a pre-trained classifier trained on natural images and to compare $\hat{P}$ and $\hat{Q}$ at a feature level. Intuitively, in this feature space the samples should be

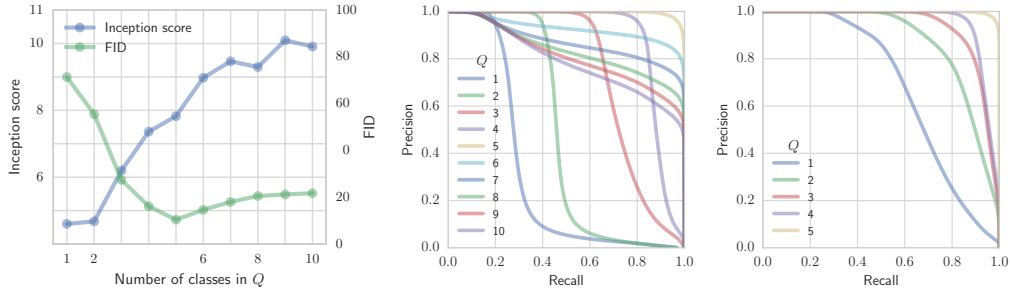

Figure 5: Left: IS and FID as we remove and add classes of CIFAR-10. IS generally only increases, while FID is sensitive to both the addition and removal of classes. However, it cannot distinguish between the two failure cases of inventing or dropping modes. Middle: Resulting PRD curves for the same experiment. As expected, adding modes leads to a loss in precision ($Q_6$–$Q_{10}$), while dropping modes leads to a loss in recall ($Q_1$–$Q_4$). As an example consider $Q_4$ and $Q_6$ which have similar FID, but strikingly different PRD curves. The same behavior can be observed for the task of text generation, as displayed on the plot on the right. For this experiment, we set $P$ to contain samples from all classes so the PRD curves demonstrate the increase in recall as we increase the number of classes in $Q$.

compared based on statistical regularities in the images rather than random artifacts resulting from the generative process [17, 19].

Following this line of work, we first use a pre-trained Inception network to embed the samples (i.e. using the *Pool3* layer [9]). We then cluster the union of $\hat{P}$ and $\hat{Q}$ in this feature space using mini-batch k-means with $k = 20$ [21]. Intuitively, we reduce the problem to a one dimensional problem where the histogram over the cluster assignments can be meaningfully compared. Hence, failing to produce samples from a cluster with many samples from the true distribution will hurt recall, and producing samples in clusters without many real samples will hurt precision. As the clustering algorithm is randomized, we run the procedure several times and average over the PRD curves. We note that such a clustering is meaningful as shown in Figure 9 in the appendix and that it can be efficiently scaled to very large sample sizes [1, 2].

We stress that from the point of view of the proposed algorithm, only a meaningful embedding is required. As such, the algorithm can be applied to various data modalities. In particular, we show in Section 4.1 that besides image data the algorithm can be applied to a text generation task.

## 4.1 Adding and dropping modes from the target distribution

Mode collapse or mode dropping is a major challenge in GANs [8, 20]. Due to the symmetry of commonly used metrics with respect to precision and recall, the only way to assess whether the model is producing low-quality images or dropping modes is by visual inspection. In stark contrast, the proposed metric can quantitatively disentangle these effects which we empirically demonstrate.

We consider three data sets commonly used in the GAN literature: MNIST [15], Fashion-MNIST [25], and CIFAR-10 [13]. These data sets are labeled and consist of 10 balanced classes. To show the sensitivity of the proposed measure to mode dropping and mode inventing, we first fix $\hat{P}$ to contain samples from the first 5 classes in the respective test set. Then, for a fixed $i = 1, \ldots, 10$, we generate a set $\hat{Q}_i$, which consists of samples from the first $i$ classes from the training set. As $i$ increases, $\hat{Q}_i$ covers an increasing number of classes from $\hat{P}$ which should result in higher recall. As we increase $i$ beyond 5, $\hat{Q}_i$ includes samples from an increasing number of classes that are not present in $\hat{P}$ which should result in a loss in precision, but not in recall as the other classes are already covered. Finally, the set $\hat{Q}_5$ covers the same classes as $\hat{P}$, so it should have high precision and high recall.

Figure 5 (left) shows the IS and FID for the CIFAR-10 data set (results on the other data sets are shown in Figure 11 in the appendix). Since the IS is not computed w.r.t. a reference distribution, it is invariant to the choice of $\hat{P}$, so as we add classes to $\hat{Q}_i$, the IS increases. The FID decreases as we add more classes until $\hat{Q}_5$ before it starts to increase as we add spurious modes. Critically, FID fails to distinguish the cases of mode dropping and mode inventing: $\hat{Q}_4$ and $\hat{Q}_6$ share similar FIDs. In contrast, Figure 5 (middle) shows our PRD curves as we vary the number of classes in $\hat{Q}_i$. Adding correct modes leads to an increase in recall, while adding fake modes leads to a loss of precision.

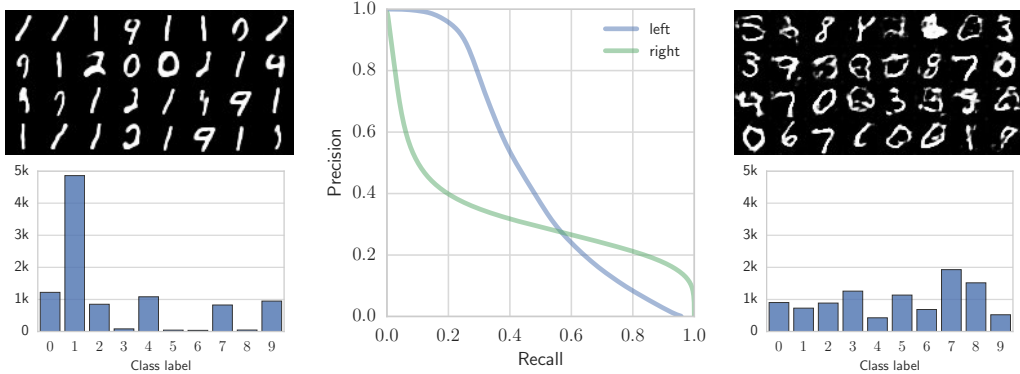

Figure 6: Comparing two GANs trained on MNIST which both achieve an FID of 49. The model on the left seems to produce high-quality samples of only a subset of digits. On the other hand, the model on the right generates low-quality samples of all digits. The histograms showing the corresponding class distributions based on a trained MNIST classifier confirm this observation. At the same time, the classifier is more confident which indicates different levels of precision (96.7% for the model on the left compared to 88.6% for the model on the right). Finally, we note that the proposed PRD algorithm does not require labeled data, as opposed to the IS which further needs a classifier that was trained on the respective data set.

We also apply the proposed approach on text data as shown in Figure 5 (right). In particular, we use the MultiNLI corpus of crowd-sourced sentence pairs annotated with topic and textual entailment information [23]. After discarding the entailment label, we collect all unique sentences for the same topic. Following [6], we embed these sentences using a BiLSTM with 2048 cells in each direction and max pooling, leading to a 4096-dimensional embedding [7]. We consider 5 classes from this data set and fix $\hat{P}$ to contain samples from all classes to measure the loss in recall for different $Q_i$. Figure 5 (right) curves successfully demonstrate the sensitivity of recall to mode dropping.

## 4.2 Assessing class imbalances for GANs

In this section we analyze the effect of class imbalance on the PRD curves. Figure 6 shows a pair of GANs trained on MNIST which have virtually the same FID, but very different PRD curves. The model on the left generates a subset of the digits of high quality, while the model on the right seems to generate all digits, but each has low quality. We can naturally interpret this difference via the PRD curves: For a desired recall level of less than ~0.6, the model on the left enjoys higher precision – it generates several digits of high quality. If, however, one desires a recall higher than ~0.6, the model on the right enjoys higher precision as it covers all digits. To confirm this, we train an MNIST classifier on the embedding of $\hat{P}$ with the ground truth labels and plot the distribution of the predicted classes for both models. The histograms clearly show that the model on the left failed to generate all classes (loss in recall), while the model on the right is producing a more balanced distribution over all classes (high recall). At the same time, the classifier has an average *confidence*[3] of 96.7% on the model on the left compared to 88.6% on the model on the right, indicating that the sample quality of the former is higher. This aligns very well with the PRD plots: samples on the left have high quality but are not diverse in contrast to the samples on the right which are diverse but have low quality.

This analysis reveals a connection to IS which is based on the premise that the conditional label distribution $p(y|x)$ should have low entropy, while the marginal $p(y) = \int p(y|x = G(z))dz$ should have high entropy. To further analyze the relationship between the proposed approach and PRD curves, we plot $p(y|x)$ against precision and $p(y)$ against recall in Figure 10 in the appendix. The results over a large number of GANs and VAEs show a large Spearman correlation of -0.83 for precision and 0.89 for recall. We however stress two key differences between the approaches: Firstly, to compute the quantities in IS one needs a classifier and a labeled data set in contrast to the proposed PRD metric which can be applied on unlabeled data. Secondly, IS only captures losses in recall w.r.t. classes, while our metric measures more fine-grained recall losses (see Figure 8 in the appendix).

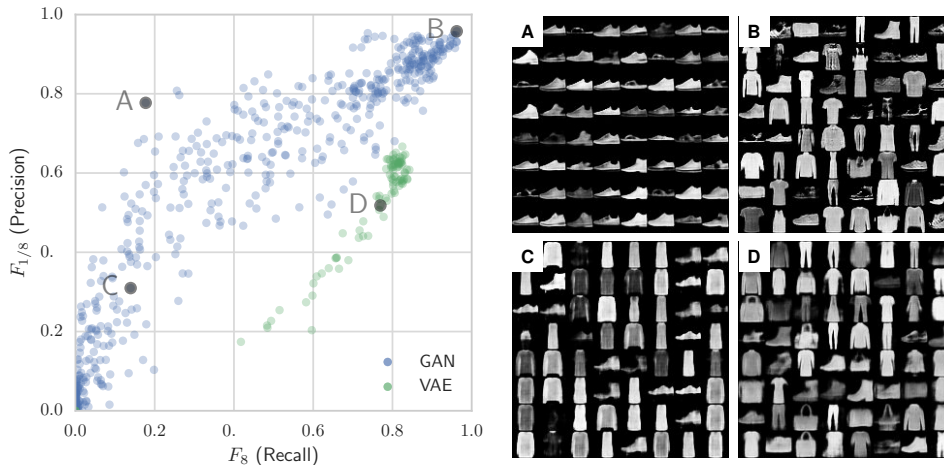

Figure 7: $F_{1/8}$ vs $F_8$ scores for a large number of GANs and VAEs on the Fashion-MNIST data set. For each model, we plot the maximum $F_{1/8}$ and $F_8$ scores to show the trade-off between precision and recall. VAEs generally achieve lower precision and/or higher recall than GANs which matches the folklore that VAEs often produce samples of lower quality while being less prone to mode collapse. On the right we show samples from four models which correspond to various success/failure modes: (A) high precision, low recall, (B) high precision, high recall, (C) low precision, low recall, and (D) low precision, high recall.

## 4.3 Application to GANs and VAEs

We evaluate the precision and recall of 7 GAN types and the VAE with 100 hyperparameter settings each as provided by [18]. In order to visualize this vast quantity of models, one needs to summarize the PRD curves. A natural idea is to compute the maximum $F_1$ score, which corresponds to the harmonic mean between precision and recall as a single-number summary. This idea is fundamentally flawed as $F_1$ is symmetric. However, its generalization, defined as $F_\beta = (1 + \beta^2)\frac{p \cdot r}{(\beta^2 p) + r}$, provides a way to quantify the relative importance of precision and recall: $\beta > 1$ weighs recall higher than precision, whereas $\beta < 1$ weighs precision higher than recall. As a result, we propose to distill each PRD curve into a pair of values: $F_\beta$ and $F_{1/\beta}$.

Figure 7 compares the maximum $F_8$ with the maximum $F_{1/8}$ for these models on the Fashion-MNIST data set. We choose $\beta = 8$ as it offers a good insight into the bias towards precision versus recall. Since $F_8$ weighs recall higher than precision and $F_{1/8}$ does the opposite, models with higher recall than precision will lie below the diagonal $F_8 = F_{1/8}$ and models with higher precision than recall will lie above. To our knowledge, this is the first metric which confirms the folklore that VAEs are biased towards higher recall, but may suffer from precision issues (e.g., due to blurring effects), at least on this data set. On the right, we show samples from four models on the extreme ends of the plot for all combinations of high and low precision and recall. We have included similar plots on the MNIST, CIFAR-10 and CelebA data sets in the appendix.

## 5 Conclusion

Quantitatively evaluating generative models is a challenging task of paramount importance. In this work we show that one-dimensional scores are not sufficient to capture different failure cases of current state-of-the-art generative models. As an alternative, we propose a novel notion of precision and recall for distributions and prove that both notions are theoretically sound and have desirable properties. We then connect these notions to total variation distance as well as FID and IS and we develop an efficient algorithm that can be readily applied to evaluate deep generative models based on samples. We investigate the properties of the proposed algorithm on real-world data sets, including image and text generation, and show that it captures the precision and recall of generative models. Finally, we find empirical evidence supporting the folklore that VAEs produce samples of lower quality, while being less prone to mode collapse than GANs.

## Footnotes

[2]For a distribution $P$ defined on a finite state space $\Omega$, we define $\mathrm{supp}(P) = \{\omega \in \Omega \mid P(\omega) > 0\}$.

[3]We denote the output of the classifier for its highest value at the softmax layer as confidence. The intuition is that higher values signify higher confidence of the model for the given label.

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
