[Supplementary Material · supplementary.pdf]

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

# A Proofs

We first show the following auxiliary result and then prove Theorems 1 and 2.

**Lemma 1.** *Let $P$ and $Q$ be probability distributions defined on a finite state space $\Omega$. Let $\alpha \in (0,1]$ and $\beta \in (0,1]$. Then, $(\alpha, \beta) \in \mathrm{PRD}(Q, P)$ if and only if there exists a distribution $\mu$ such that for all $\omega \in \Omega$*

$$P(\omega) \geq \beta\mu(\omega) \quad and \quad Q(\omega) \geq \alpha\mu(\omega). \tag{5}$$

*Proof.* If $(\alpha, \beta) \in \mathrm{PRD}(Q, P)$, then (3) and the non-negativity of $\nu_P$ and $\nu_Q$ directly imply (5) for the same choice of $\mu$. Conversely, if (5) holds for a distribution $\mu$, we may define the distributions

$$\nu_P(\omega) = \frac{P(\omega) - \beta\mu(\omega)}{1 - \beta} \quad and \quad \nu_Q(\omega) = \frac{Q(\omega) - \alpha\mu(\omega)}{1 - \alpha}.$$

By definition $\alpha$, $\beta$, $\mu$, $\nu_P$ and $\nu_Q$ satisfy (3) in Definition 1 which implies $(\alpha, \beta) \in \mathrm{PRD}(Q, P)$. $\square$

## A.1 Proof of Theorem 1

*Proof.* We show each of the properties independently:

*(i) Equality*: If $(1,1) \in \mathrm{PRD}(Q, P)$, then we have by Definition 1 that $P = \mu$ and $Q = \mu$ which implies $P = Q$ as claimed. Conversely, if $P = Q$, Definition 1 is satisfied for $\alpha = \beta = 1$ by choosing $\mu = \nu_P = \nu_Q = P$. Hence, $(1,1) \in \mathrm{PRD}(Q, P)$ as claimed.

*(ii) Disjoint support*: We show both directions of the claim by contraposition, i.e., we show $\mathrm{supp}(P) \cap \mathrm{supp}(Q) \neq \emptyset \Leftrightarrow \mathrm{PRD}(Q, P) \supset \{(0,0)\}$. Consider an arbitrary $\omega \in \mathrm{supp}(P) \cap \mathrm{supp}(Q)$. Then, by definition we have $P(\omega) > 0$ and $Q(\omega) > 0$. Let $\mu$ be defined as the distribution with $\mu(\omega) = 1$ and $\mu(\omega') = 0$ for all $\omega' \in \Omega \setminus \{\omega\}$. Clearly, it holds that $P(\omega) \geq P(\omega)\mu(\omega)$ and $Q(\omega) \geq Q(\omega)\mu(\omega)$ for all $\omega \in \Omega$. Hence, by Lemma 1, we have $(Q(\omega), P(\omega)) \in \mathrm{PRD}(Q, P)$ which implies that $\mathrm{PRD}(Q, P) \supset \{(0,0)\}$ as claimed. Conversely, $\mathrm{PRD}(Q, P) \supset \{(0,0)\}$ implies by Lemma 1 that there exist $\alpha \in (0,1]$ and $\beta \in (0,1]$ as well as a distribution $\mu$ satisfying (5). Let $\omega \in \mathrm{supp}(\mu)$ which implies $\mu(\omega) > 0$ and thus by (5) also $P(\omega) > 0$ and $Q(\omega) > 0$. Hence, $\omega$ is both in the support of $P$ and $Q$ which implies $\mathrm{supp}(P) \cap \mathrm{supp}(Q) \neq \emptyset$ as claimed.

*(iii) Maximum precision*: If $(\alpha, \beta) \in \mathrm{PRD}(Q, P)$, then by Lemma 1 there exists a distribution $\mu$ such that for all $\omega \in \Omega$ we have $P(\omega) \geq \beta\mu(\omega)$ and $Q(\omega) \geq \alpha\mu(\omega)$. $P(\omega) \geq \beta\mu(\omega)$ implies $\mathrm{supp}(\mu) \subseteq \mathrm{supp}(P)$ and hence $\sum_{\omega \in \mathrm{supp}(P)} \mu(\omega) = 1$. Together with $Q(\omega) \geq \alpha\mu(\omega)$, this yields $Q(\mathrm{supp}(P)) = \sum_{\omega \in \mathrm{supp}(P)} Q(\omega) \geq \alpha \sum_{\omega \in \mathrm{supp}(P)} \mu(\omega) = \alpha$ which implies $\alpha \leq Q(\mathrm{supp}(P))$ for all $(\alpha, \beta) \in \mathrm{PRD}(Q, P)$.

To prove the claim, we next show that there exists $(\alpha, \beta) \in \mathrm{PRD}(Q, P)$ with $\alpha = Q(\mathrm{supp}(P))$. Let $S = \mathrm{supp}(P) \cap \mathrm{supp}(Q)$. If $S = \emptyset$, then $\alpha = Q(\mathrm{supp}(P)) = 0$ and $(0,0) \in \mathrm{PRD}(Q, P)$ by Definition 2 as claimed. For the case $S \neq \emptyset$, let $\beta = \min_{\omega \in S} \frac{P(\omega)Q(S)}{Q(\omega)}$. By definition of $S$, we have $\beta > 0$. Furthermore, $\beta \leq P(S) \leq 1$ since $\frac{P(\omega)}{P(S)} \leq \frac{Q(\omega)}{Q(S)}$ for at least one $\omega \in S$. Consider the distribution $\mu$ where $\mu(\omega) = \frac{Q(\omega)}{Q(S)}$ for all $\omega \in S$ and $\mu(\omega) = 0$ for $\omega \in \Omega \setminus S$. By construction, $\mu$ satisfies (5) in Lemma 1 and hence $(\alpha, \beta) \in \mathrm{PRD}(Q, P)$ as claimed.

*(iv) Maximum recall*: This follows directly from applying Property (vi) to Property (iii).

*(v) Monotonicity*: If $(\alpha, \beta) \in \mathrm{PRD}(Q, P)$, then by Lemma 1 there exists a distribution $\mu$ such that for all $\omega \in \Omega$ we have that $P(\omega) \geq \beta\mu(\omega)$ and $Q(\omega) \geq \alpha\mu(\omega)$. For $\alpha' \in (0, \alpha]$ and $\beta' \in (0, \beta]$, it follows that $P(\omega) \geq \beta'\mu(\omega)$ and $Q(\omega) \geq \alpha'\mu(\omega)$ for all $\omega \in \Omega$. By Lemma 1 this implies $(\alpha', \beta') \in \mathrm{PRD}(Q, P)$ as claimed.

*(vi) Duality*: This follows directly from switching $\alpha$ with $\beta$, $P$ with $Q$ and $\nu_P$ with $\nu_Q$ in Definition 1.

$\square$

## A.2 Proof of Theorem 2

*Proof.* We first show that $\mathrm{PRD}(Q,P) \subseteq \{(\theta\alpha(\lambda), \theta\beta(\lambda)) \mid \lambda \in (0,\infty), \theta \in [0,1]\}$. We consider an arbitrary element $(\alpha',\beta') \in \mathrm{PRD}(Q,P)$ and show that $(\alpha',\beta') = (\theta\alpha(\lambda), \theta\beta(\lambda))$ for some $\lambda \in (0,\infty)$ and $\theta \in [0,1]$. For the case $(\alpha',\beta') = (0,0)$, the result holds trivially for the choice of $\lambda = 1$ and $\theta = 0$. For the case $(\alpha',\beta') \neq (0,0)$, we choose $\lambda = \frac{\alpha'}{\beta'}$ and $\theta = \frac{\beta'}{\beta(\lambda)}$. Since $\alpha(\lambda) = \lambda\beta(\lambda)$ by definition, this implies $(\alpha',\beta') = (\theta\alpha(\lambda), \theta\beta(\lambda))$ as required. Furthermore, $\lambda \in (0,\infty)$ since by Definitions 1 and 2 $\alpha' > 0$ if and only if $\beta' > 0$. Similarly, we show that $\theta \in [0,1]$: By Lemma 1 there exists a distribution $\mu$ such that $\beta'\mu(\omega) \leq P(\omega)$ and $\alpha'\mu(\omega) \leq Q(\omega)$ for all $\omega \in \Omega$. This implies that $\beta'\mu(\omega) \leq \frac{Q(\omega)}{\lambda}$ and thus $\beta'\mu(\omega) \leq \min\left(P(\omega), \frac{Q(\omega)}{\lambda}\right)$ for all $\omega \in \Omega$. Summing over all $\omega \in \Omega$, we obtain $\beta' \leq \sum_{\omega \in \Omega} \min\left(P(\omega), \frac{Q(\omega)}{\lambda}\right) = \beta(\lambda)$ which implies $\theta \in [0,1]$.

Finally, we show that $\mathrm{PRD}(Q,P) \supseteq \{(\theta\alpha(\lambda), \theta\beta(\lambda)) \mid \lambda \in (0,\infty), \theta \in [0,1]\}$. Consider arbitrary $\lambda \in (0,\infty)$ and $\theta \in [0,1]$. If $\beta(\lambda) = 0$, the claim holds trivially since $(0,0) \in \mathrm{PRD}(Q,P)$. Otherwise, define the distribution $\mu$ by $\mu(\omega) = \min\left(P(\omega), \frac{Q(\omega)}{\lambda}\right)/\beta(\lambda)$ for all $\omega \in \Omega$. By definition, $\beta(\lambda)\mu(\omega) \leq \min\left(P(\omega), \frac{Q(\omega)}{\lambda}\right) \leq P(\omega)$ for all $\omega \in \Omega$. Similarly, $\alpha(\lambda)\mu(\omega) \leq \min(\lambda P(\omega), Q(\omega)) \leq Q(\omega)$ for all $\omega \in \Omega$ since $\alpha(\lambda) = \lambda\beta(\lambda)$. Because $\theta \in [0,1]$, this implies $\theta\beta(\lambda)\mu(\omega) \leq P(\omega)$ and $\theta\alpha(\lambda)\mu(\omega) \leq Q(\omega)$ for all $\omega \in \Omega$. Hence, by Lemma 1, $(\theta\alpha(\lambda), \theta\beta(\lambda)) \in \mathrm{PRD}(Q,P)$ for all $\lambda \in (0,\infty)$ and $\theta \in [0,1]$ as claimed. $\square$

# B  Further figures

Figure 8: Comparing a pair of GANs on MNIST which have both collapsed to producing 1's. An analysis with a trained classifier as in Section 4.2 comes to the same conclusion for both models, namely, that they have collapsed to producing 1's only. However, the PRD curve shows that the model on the right has a slightly higher recall. This is indeed correct: while the model on the left is producing straight 1's only, the model on the right is producing some more varied shapes such as tilted 1's.

Real images                                    Generated images

Figure 9: Clustering the real and generated samples from a GAN in feature space (10 cluster centers for visualization) yields the clusters above for the data sets MNIST, Fashion-MNIST, CIFAR-10 and CelebA. Although the GAN samples are not perfect, they are clustered in a meaningful way.

Figure 10: Comparing our unsupervised $F_{1/8}$ and $F_8$ measures with the supervised measures $P(y|x)$ and $P(y)$ similar to the IS (for a definition of $F_\beta$, see Section 4.3). Each circle represents a trained generative model (GAN or VAE) on the MNIST data set. The values show a fairly high correlation with a Spearman rank correlation coefficient of -0.83 on the left and 0.89 on the right.

Figure 11: Corresponding plots as in Figure 5 for the data sets MNIST (top) and Fashion-MNIST (bottom).

Figure 12: Corresponding plots as in Figure 7 for the data sets MNIST (top), CIFAR-10 (middle) and CelebA (bottom).