[Reviews · NeurIPS 2018]

Reviewer 1



This paper contributes some original thinkings on how to assess the quality of a generative model. The new evaluation metric, as defined by the distributional precision and recall statistics (PRD), overcomes a major drawback from prior-arts: that evaluation scores are almost exclusively scalar metrics. The author(s) attributes intuitive explanations to this new metric and experimentally reached a conclusion that it is able to disentangle the quality from the coverage, two critical aspects wrt the quality of a learned synthetic sampler. An efficient algorithm is also described and theoretically justified. The quality of this work seems okay, yet I am prone to a neutral-to-negative rating. This is mainly because I am not sure I fully understand the technical details from the paper after a few attempts, and I have doubts about its value to the future development of improved generative models. I will reconsider my decision if the author(s) can properly address my concerns detailed below. * While I strongly concur with the fact that a comprehensive set of indicators are needed to properly evaluate the quality of a generative model, it is not so obvious to me why a curve-type metric like PRD proposed in the current study is more preferable. Such metrics can be more open to (subjective) interpretations, which I found undesirable. Because this might add to the confusion of what makes a good generative model. * Overload of symbol \bar{\apha} and \bar{\beta}. * When doing a comparison involving more than 3 distributions, shouldn't we use the pooled samples from all the models (and data) to construct the local bins? (As oppose to pair-wise matching for the evaluation of PRD score, which I assume is what the author(s) did in the current experiments). * The author(s) kind of implies there is this trade-off between precision and recall, at least in practice. I need more proof on this point. * As also mentioned by the author(s), KNN may not be a reliable tool as the cluster assignment might be arbitrary. * Fig. 7 left, adding marginal histograms might serve to better interpret the observation. * The experimental results in the main text seem a little seem somewhat weak, as all experiments are only based on simple binary/gray-scale images. More extensive set of experiments, like some representative toy models and more sophisticated image datasets, should be considered in the main text. * This paper does not provide any insights on how to improve generative modeling wrt the novel metric discussed. And this is a major drawback.

Reviewer 2



The authors propose a new way of evaluating generative models using a generalization of precision and recall to generative model. Intuitively, a model has high recall if it captures many modes of the target distribution; a model has high precision if it captures these models well. The authors propose describing the quality of a generative model via a precision-recall curve. The shape of this curve provides information that traditional measures like the Inception score miss (e.g. are we covering many modes well or just one mode). I think the paper studies a very important topic in generative modeling and provides a way of measuring the quality of generative samples in a way that other models cannot. This allows formally quantifying several crucial issues in generative modeling such as mode collapse, which were only studies qualitatively until now. The paper is also very well written and offers a very thorough discussion of model quality. I suspect this metric will be used in practice. Comments ------------- - I am still a bit confused as to why the proposed method looks at a tradeoff between precision and recall. In binary classification, there is truly a tradeoff because by varying the threshold outputs a different set of probabilities. But here, the model always produces the same samples; its not like we can explicitly construct the \mu. Hence, two distinct scalars measuring precision and recall (something like a snapshot along that curve) would make more sense to me. It would be great to have a more thorough discussion of this in the paper. - All the (elegant) theory assumes that P, Q are over discrete spaces. In practice, the input space is quantized. What is lost during this approximation? How does the number of k-means clusters affect the results? How do we choose k? I don't see a good discussion in the paper. - Is there a reason why the main results for real generative models are focused on MNIST and Fashion-MNIST? There some plots for Cifar10 and CelebA in the appendix, but there is clearly much more focus on MNIST. This makes the argument seem a bit weaker. - What is the importance of the feature embedding in which you run k-means? How do I find such features for non-image data? How does the choice of features affect the quality of the results? - Is it possible to summarize the curve by a single value (like the area under the curve)? Does it give useful insights?

Reviewer 3



The paper proposes a performance metric for generative model evaluation. Different to Frechet Inception Distance (FID) and Inception Score (IS), which only give a single performance number, the proposed metric gives a precision--recall curve where the definition of precision and recall are tailored for comparing distributions. The precision measures the quality of generated samples, while the recall checks the number of modes in the original distribution that is covered by the learned distribution. This evaluation metric appears to be more informative than FID and IS. Strength: - The paper studies an important problem in generative model evaluation. A good evaluation metric can help the community better evaluate generative models. - The idea of using precision-and-recall curve for evaluating generative model is novel. The idea is well-motivated and justified in the paper. The derivation of a simple approach to compute the PRD curve is also quite interesting. Weakness: - The fact that a closed-form solution for computing the PRD curve does not exist is a bit disappointing. Since the performance number depends on the K-mean clustering, which can give different results for different random seeds, it will be great if the authors could show the variation of the PRD curves in addition to the average. Also, the authors should analyze the impact of number of clusters used in the K-mean algorithm. Is the performance metric sensitive to the choice of number of K-mean centers? What would be a reasonable number of clusters for a large dataset such as the ImageNet dataset.